# An Animation- Versus Text-Based Computer-Tailored Game Intervention to Prevent Alcohol Consumption and Binge Drinking in Adolescents: Study Protocol

**DOI:** 10.3390/ijerph18199978

**Published:** 2021-09-23

**Authors:** Marta Lima-Serrano, Pablo Fernández-León, Liesbeth Mercken, José Manuel Martínez-Montilla, Hein de Vries

**Affiliations:** 1Department of Nursing, School of Nursing, Physiotherapy and Podiatry, University of Seville, 41009 Seville, Spain; mlima@us.es; 2Department of Health Promotion, CAPHRI Care and Public Health Research Institute, Maastricht University, 6229 HA Maastricht, The Netherlands; liesbeth.mercken@ou.nl (L.M.); hein.devries@maastrichtuniversity.nl (H.d.V.); 3Department of Health Psychology, Open University Heerlen, 6419 AT Heerlen, The Netherlands; 4Spanish Red Cross Nursing School, University of Seville, 41009 Seville, Spain; jose.martinezm@cruzroja.es

**Keywords:** alcohol use, binge drinking, cluster-randomized controlled trial, adolescents, web-based interventions, computer-tailoring, animation

## Abstract

The purpose of this paper is to describe the protocol for the design, implementation, and evaluation of an animation- versus text-based computer tailoring game intervention aimed at preventing alcohol consumption and binge drinking (BD) in adolescents. A cluster-randomized controlled trial (CRCT) is carried out in students aged 14–19 enrolled in 24 high schools from Andalusia (Spain), which are randomized either to experimental (EC-1, EC-2) or waiting-list control conditions (CC). EC-1 receives an online intervention (Alerta Alcohol) with personalized health advice, using textual feedback and several gamification techniques. EC-2 receives an improved version (Alerta Alcohol 2.0) using animated videos and new gamification strategies. Both programs consist of nine sessions (seven taking place at high school and two at home): session 1 or baseline, sessions 2 and 3 that provide tailored advice based on the I-Change Model; sessions 4, 5, 7, and 8 are booster sessions, and sessions 6 and 9 are follow-up questionnaires at six and twelve months. The CC completes the baseline and the evaluation questionnaires. The primary outcome is BD within 30 days before post-test evaluations, and as secondary outcomes we assess other patterns of alcohol use. The findings should help the development of future alcohol drinking prevention interventions in adolescents.

## 1. Introduction

Alcohol use and misuse is one of the leading risk factors for global population health and it results in around 3 million deaths (5.3% of all deaths) worldwide [1]. Alcohol can harm almost any system or organ of the body. It is related to more than 60 different disorders with short- and long-term consequences and also associated with an increased risk of third-party harm [2]. Nevertheless, alcohol consumption has become part of the social life, resulting in a specific “drinking culture” that may also differ from region to region [3].

### 1.1. Patterns of Alcohol Consumption in the Adolescent Population

Binge drinking (BD) is a problematic pattern of alcohol consumption especially frequent in the adolescent population [4]. There is a lack of consensus regarding the operational definition of BD concerning the number of standard drink units (SDUs) or standard glasses of alcohol consumed per occasion (the value of the Spanish SDU is 10 g of alcohol), the measures of blood alcohol concentration, and the frequency or duration of episodes [5]. However, the criterion most widely accepted by the international scientific community is the consumption of five or more SDUs by men and four or more by women in a short space of time or during a single occasion [4,5,6]. In addition, many researchers examined another problematic pattern that has been called excessive drinking, heavy episodic drinking, or high-intensity drinking (HID), which is described as twice the typical BD threshold (i.e., 10 or more SDUs on at least one drinking occasion) [7,8,9,10] or twice the typical gender-specific BD threshold (i.e., 10 or more SDUs by men and eight or more by women) [10].

BD and HID have been associated with detrimental long- and short-term consequences, such as traffic accidents, violence, homicide, suicide, early and risky sexual contact, academic or occupational failure, mental illnesses, and delinquency [11,12,13]. Moreover, HID is of particular concern because of the more serious adverse consequences including alcohol poisoning, passing out, and blacking out [13]. According to the latest Spanish epidemiological data from the survey on the Use of Drugs in Secondary Education (ESTUDES 2018/2019) [14], a total of 75.9% of adolescents between 14 and 18 years of age drank alcohol in the last 12 months and 32.3% reported BD in the last 30 days. Furthermore, in recent years it observes an increasing trend in the proportion of students who got drunk and the students who reported BD in the last 30 days [14,15]. In Andalusia (southern Spain), in the survey The Andalusian Population Facing Drugs [16], a total of 51% of adolescents aged 14–20 reported BD in the last 30 days. Another study mentioned a prevalence of BD of 39.1% in adolescents aged 15–19 [9]. Not one of the previous national or regional surveys shows data on HID. However, Best et al. [7] found a prevalence of 32% of adolescents (aged 14–16-year-old) who drank 10 or more glasses of alcohol on any drinking occasion in the UK. Other authors found a HID prevalence of 1% to 6.7% in adolescents aged 14–20 in the USA, 9.2% in adolescents aged 15–19 in The Netherlands, and 1.3% in adolescents aged 15–19 in Andalusia (Spain) [8,9,17]. These figures highlight the importance of preventing both BD and HID. The study described below focuses on the development and testing of a computer-tailored eHealth intervention aimed at preventing BD and HID and is grounded in the integration of social cognitive theories, as outlined by the I-Change Model (2017) and building on earlier work [8,9].

### 1.2. The I-Change Model

The I-Change Model [18] focuses behavioral change into three phases: pre-motivational (awareness), motivational (motivation), and post-motivational (action), which is moderated by information factors, i.e., personal, message, channel and source, and preceding factors, i.e., biological, psychological, behavioral, and environmental (Figure 1). The model postulates that pre-motivational factors (i.e., knowledge, risk perceptions, cues to action and cognizance) play a distal role, meaning that firstly there is a need for self-awareness as regards behavior and the need for behavioral change, in order to create motivation. Subsequently, sufficient awareness of behavioral change is necessary. Attitudes, social influences, and self-efficacy determine the intention of a person to change a certain behavior. Once a high intention to change behavior is present, research shows that this does not automatically warrant change [19]. Individuals can facilitate behavioral change by making preparatory and coping plans and applying them in practice. In this phase, again, self-efficacy plays a major role in carrying out action plans. This model has been used to predict different behaviors related to alcohol consumption in Dutch and Spanish adolescents and parents, as well as in Dutch midwives, pregnant women and their partners [12,20,21], and has also been employed for the prevention of alcohol consumption through computer-tailoring technology (CTT) in adults [22,23,24] and adolescent populations [8,9].

### 1.3. Computer-Tailoring Technology

CTT is a prevention technology that consists of providing participants personalized advice based on unique answers given on a web-based assessment [25]. Two important benefits must be noted regarding this kind of technology: on one hand, accessibility to the web by not having spatiotemporal limits and, on the other hand, the capacity to generate personalized messages based on participants’ motivational characteristics which facilitate attracting the attention of individuals and the processing of information, and thus changing behavior [26,27,28]. Furthermore, because of the COVID-19 pandemic digital health interventions for prevention are currently particularly recommended for addressing the challenges of the virus (and potential future pandemics) resulting in more online communication [29,30,31]. In Spain, this technology has the potential to reach a large part of the adolescent population. For example, 99.8% of young people between 16–24 are internet users, with hardly any differences in terms of gender and/or social class [32]. Moreover, the advantage of increased internet-based programs at educational centers makes schools very suitable places for implementing these interventions [8,9,26,33,34,35,36].

Jander et al. [8,26] developed a dynamically tailored internet-based intervention in the context of a tailored game, for the prevention of BD in Dutch adolescents, aiming to change the motivational factors and the actual occurrence of BD. In this intervention, the adolescents received personalized information on their consumption behavior, as well as messages aimed at its prevention. It resulted effective in reducing BD among adolescents aged 15 years (*p* = 0.03) and those aged 16 years when they participated in at least two intervention sessions (*p* = 0.04). Follow-up assessment of alcohol use took place four months later the baseline [8]. In addition, from both the health care and the societal perspective, the intervention was more cost-effective in reducing the number of BD occasions per month for older adolescents (aged 17–19) than for those who were under 17 years old [37].

In Spain, Lima et al. [9,34] implemented and evaluated a web-based computer-tailored (CT) intervention (Alerta Alcohol), which is a cultural adaptation of the Dutch program [26], using short stories and personalized text messages through a website. Alerta Alcohol incorporated some gamification elements such as offering a challenge, presenting different stories, and the use of avatars therein [9]. The effectiveness of the program to reduce the probability of BD at the four-month follow-up was statistically significant, such as a higher adherence to the program (a higher number of sessions filled), which showed a reduction in the number of occasions of BD [38]. Alerta Alcohol also appeared to be a cost-effective way to prevent BD in terms of reducing the number of BD occasions and of increasing quality-adjusted life-years among adolescents and especially for specific subgroups of this population [39]. In addition, the program resulted in a significant reduction of heavy episodic drinking (operationally defined as HID in this study) [9].

### 1.4. The Current Study

The aim of this paper is to describe the protocol for the design, implementation, and evaluation of an animation (Alerta Alcohol 2.0)- versus text-based (Alerta Alcohol) CT game intervention aimed to prevent alcohol consumption and BD in Spanish adolescents.

There is evidence that suggests that gamification tactics may help to increase impact in health behavior through an increase in user’ attention and engagement [40,41]. Moreover, gamification techniques such as using rewards or displaying user’ progress show benefits specifically in e-Health interventions. Using rewards (e.g., points or achievement badges) is a core gamification strategy that accomplishes the requested tasks, and it is used to promote competition among users [40,41]. Users’ progress is mostly fuelled by the number of rewards collected, which are then used to provide a ranking on leaderboards as another gamification element [41]. Moreover, another way of improving users’ experience is to adapt the mode of delivery (text messages, video messages, or animations) to the needs of the target population, as this has the potential to enhance the effects of CT interventions [42]. The use of video-delivered tailored information that makes use of words (i.e., spoken text) and graphics (i.e., animations) has been shown to result in more extensive information processing than text-based interventions that rely solely on written words [43]. Animations may be better able to support the creation of an adequate mental representation, enhancing learning and understanding, when compared to video illustrations [44,45,46]. Other studies show that spoken messages can be better recalled and induce more positive attitudes compared to written texts among people with low health literacy, and do not negatively influence high health literate target groups [46,47]. Furthermore, in the prevention of drug dependence, animated-based CT interventions have shown to result in higher quit rates in smoking than text-based CT interventions [48].

The purpose of this paper is to describe the protocol for the design, implementation, and evaluation of an animation (Alerta Alcohol 2.0)- versus text-based (Alerta Alcohol) CT game intervention aimed at preventing alcohol consumption and BD in Spanish adolescents.

## 2. Materials and Methods

### 2.1. Study Design

A three-arm cluster-randomized controlled trial (CRCT) is designed, with two experimental conditions EC-1 (Alerta Alcohol text-based game), EC-2 (Alerta Alcohol 2.0 animation-based game) and one waiting-list control condition (CC) randomized at the school level, with an initial (pre-test) evaluation and two (post-test) evaluations, performed at six and twelve months (Figure 2). The original design of the program is inspired by the study of Lima et al. [9,34], which was carried out in Spain with an equivalent objective. The research team has members with sufficient and appropriate expertise and experience in the design, development, and evaluation of the program through three funded research projects and with a production of two doctoral dissertations with international mention, in collaboration with the University of Maastricht.

The SPIRIT guideline is followed [49] and outlined in Figure 2.

### 2.2. Participants

The target group for the program consists of adolescents aged 14–19 in the public or private school system in Andalusia (Spain), enrolled in their third and fourth year of compulsory secondary education (CSE), in the first year of their high school program, and those in the first year of continuing education or vocational training (VT), which is equivalent to 9th, 10th, 11th, and 12th grades, respectively, in the USA. The criteria for inclusion in the study are having internet access at school and at home. Those with language difficulties or those who previously took part in BD prevention programs are excluded. To check the inclusion criteria, a researcher is virtually present at the pre-test.

In addition, the inclusion criteria for schools are as follows: (1) public or private secondary schools from Andalusia and (2) schools with internet access and computer, mobile, or tablet use are allowed.

### 2.3. Sample Size

To calculate the sample size, the online GRANMO tool is used (http://www.imim.cat/ofertadeserveis/software-public/granmo/) (accessed on 1 September 2018). According to the ESTUDES 2018/2019 study, in Spain the prevalence of adolescent BD within the previous 30 days is 32.3% [14]. It is estimated that the intervention reduces consumption by 10% in BD. Accepting a significant *p*-value < 0.05, the statistical power of 0.80 in a bilateral contrast, 584 subjects are required for the EC-1, 584 for the EC-2, and 584 for the CC (1752 participants) to find a statistically significant proportion difference, expected to be of 0.38 for EC-1 and EC-2, and 0.28 for CC. Following the study by Lima et al., a retention rate of about 50% has been anticipated. The ARCSINE approximation is used [26,34]. Accepting the cluster design and considering an intraclass coefficient of 0.02, we estimate a sample of approximately 35 classrooms of 25 students by condition, for a total of 105 classrooms comprising around 2625 students (875 by condition) [26].

### 2.4. Selection of the Sample

The randomization process is undertaken by two researchers from the team (PFL and MLS) using the Research Randomizer (Version 4.0) computer software (http://www.randomizer.org/) to avoid contamination (accessed on 1 September 2019). First, we randomly select at least three schools from each of the eight provinces in Andalusia. The schools are informed of the objective of the study and the sessions of the intervention. Participation by each school is confirmed by email or telephone. A formal letter and an informative folder are sent to each center.

If schools agree to participate, the inclusion criteria are checked. If they do not agree to participate, we randomize other schools in the same province until at least three schools in each province are included. In total, we contact 173 Andalusian schools.

Finally, after 24 high schools, three from each province of Andalusia, are accepted, they are randomly assigned to the EC-1, EC-2, or the CC. Within each school, all classes that meet the inclusion criteria are invited to participate in the study.

The CC schools are on a waiting list and will receive the intervention voluntarily once the study is completed. The selected schools are not blinded to their groups, since the EC group needs to schedule a total of seven sessions during school hours. The adolescents are recruited from schools through their teachers and counselors. Adolescent participants and their parents have to sign and return the informed consent form to agree to take part in this scientific study. When starting the intervention, participants are asked to visit the study website and create an account. Within their account, they select their school and are assigned to one of the conditions: EC-1, EC-2, or CC. Before starting with the baseline questionnaire, students give informed consent by checking the acceptance box on the first page of the website. If he or she does not wish to participate, or refuse to provide informed consent, he or she can select the option ‘I do not wish to participate in this study’. In this case, he or she is thanked and can leave the website. Participants can, however, also access the website at another time if they want to. All students enrolled provide active informed consent for the use of their data for scientific research and publication. Those who are underage are asked that their parents complete an informed consent form. These consents are sent by email to the counselor of each school who will be in charge of delivering them to students and parents. Finally, teachers collect and return them to the research team.

### 2.5. Intervention

The study has two different ECs that are compared to a control group. The characteristics of each EC are described below:

#### 2.5.1. Experimental Condition 1 (Alerta Alcohol Program)

The Alerta Alcohol program consists of providing textual feedback through preventive messages and personalized information about the benefits of not consuming alcohol, reducing positive attitudes towards alcohol and BD, as well as social influence and self-efficacy. Skills and action plans are encouraged to help the students reject alcohol consumption or BD. To start the intervention, students have access to a website via an electronic device (computer, mobile, or tablet). The intervention consists of six sessions. Session 1 consists of a reference questionnaire. Sessions 2–3 consist of a short story in which the main character wakes up after an evening in which he/she binge drank and does not remember what happened. The stories describe situations concerning alcohol use at home (session 2), at celebrations (session 3), and in public places (session 3) as these situations were previously identified as the riskiest situations for alcohol consumption, BD and HID [12,20]. Using these scenarios, the story is presented, and questions and tailored text messages are offered based on the I-Change Model [18,50,51]. During session 2, when the story takes place at home, the feedback focuses on providing information about the general and individual consequences of alcohol consumption and BD, as well as making the adolescent aware of the negative aspects of BD and reinforcing negative beliefs thereof in order to turn attitudes against alcohol abuse. During session 3, following the second scenario in which the story involves celebrations such as Christmas, weddings, and festivals, with the messages addressing issues related to social models, helping the adolescents to choose models considered most appropriate, encouraging them to seek support from friends and family members who do not drink alcohol in excess. The third scenario, which is also included in session 3, relates to public spaces and addresses issues linked to social norms connected to BD, like the opinion of others. It helps adolescents to deal with the perceived approval of drinking among family and friends and to choose true relationships that may help them to avoid alcohol drinking and BD. Furthermore, the questions and messages address how to resist social pressure to drink alcohol from friends or family. In addition, self-efficacy to handle these situations is evaluated in all scenarios, and specific action plans are offered depending on the answers given individually. Concerning session 4, this consists of a challenge of not drinking or at least not binge drink at an upcoming drinking event, this is a booster session in which the participant is reminded of the main health messages, and in session 5 the challenge is evaluated. Finally, session 6 consists of the evaluation (i.e., the follow-up questionnaire) of the intervention. A detailed description of the development and content of the intervention is available elsewhere [9,34].

Several gamification elements are used in Alerta Alcohol. First, participants are invited to participate in a challenge of not consuming alcohol in excess at an upcoming event. Second, the intervention consists of short stories in which the main character binge drank the night before and his or her friends talk with him or her about what happened. The stories were designed based on the results of a focus group study and were adapted to the gender of the participant [20]. Finally, participants can choose an avatar (four different, two males and two females) and the names of the characters in the stories.

Taking into account previous recommendations from the original Alerta Alcohol program [9], the following changes have been included in Alerta Alcohol in this project. First, two booster sessions, the same as the previous booster sessions but a year after the last scenario (sessions 7 and 8), and a second follow-up questionnaire (session 9) are added, totaling nine sessions. Second, four more avatars (two males and two females) are introduced, totaling eight avatars adapted to the gender, sexual orientation, and ethnicity. Third, text health messages are shortened. Fourth, images are included in the text to increase risk perception. Fifth, instead of inviting youngsters to do these at home, sessions 4 and 7 (the challenges) are presented during school time to improve adherence to the intervention.

#### 2.5.2. Experimental Condition 2 (Alerta Alcohol 2.0 Program)

The Alerta Alcohol 2.0 program is based on the Alerta Alcohol program after revision of the original design. However, several changes have been introduced.

First, the different stories and the intervention routings (the scenarios and tailored text messages) used in the Alerta Alcohol program are converted to animated videos (Figure 3), which have been developed by the Audiovisual Resources and New Technologies Secretariat (http://sav.us.es/) (accessed on 15 April 2019), a professional-level audiovisual and ICT University of Seville company. New avatars are designed, but due to this conversion to animated videos, it was decided to only elaborate five (two males and three females), which already had a name assigned and which accompany it throughout the intervention (Figure 4).

Second, new gamification strategies are added to those already used in Alerta Alcohol (i.e., offering a challenge, giving different stories, and the use of avatars in these stories). Firstly, participants receive a reward (a card) with a healthy message when some sessions are completed (Figure 5). The program has a total of nine cards. Secondly, these cards have a scoring (10 points each) that users will accumulate. Third, user position is ranked based on the accumulative score, and they can check it through the Alerta Alcohol 2.0 website (https://alcoholalerta.es/) (accessed on 18 May 2020) (Figure 6).

#### 2.5.3. Control Condition

Participants in the control condition only receive the pre-test, one post-test at six months, and the last post-test at twelve months (sessions 1, 6, and 9). They are asked to complete the same study measurements as those in the ECs, at the same time points.

### 2.6. Measurements

We use a Spanish validated version of the self-administered online questionnaire [52], which was used by Lima et al. [34] and adapted from the previous study carried out on Dutch adolescents [26]. The outcome measurements are assessed at three-time points, namely baseline and six and twelve months following baseline.

#### 2.6.1. Social-Demographic Variables

We measure the following demographic characteristics: gender (male/female), age (in years), parents’ educational level (none, primary, high school, university, don’t know), academic course (elementary school, high school, VT), religion (Catholic, Protestant/Evangelical, Muslim/Islamic, other religion, no religion), and nationality (Spanish, other).

In addition, social status is measured by the latest version of the Family Affluence Scale (FAS-III) which consists of six different questions (Does your family own a car, van, or truck? Do you have your own bedroom? How many times did you and your family travel outside Spain for a vacation last year? How many computers (including laptops and tablets, not including game consoles and smartphones) does your family own? Does your family have a dishwasher at home? How many bathrooms (rooms with a bath/shower or both) are in your home?) [53,54].

The Family Apgar Test is used to measure self-perception on familiar functional status. It consists of five questions answered by a Likert three-point scale, assessing the adaptability or resource mobilization (Are you satisfied with the help you received from your family when you have problems?), participation or cooperation (Do you talk at home about the problems you have?), development or growth (Are important family decisions discussed together at home?), resolution or capacity of spending time with a family member (Are you satisfied with the time you spend together with your family?), and affection (Do you feel that your family loves you?) [55].

#### 2.6.2. Drinking Behaviors

We assess four drinking patterns: BD, HID, weekly consumption, and any consumption.

BD is assessed using an open-ended question on how many BD occasions they participated in during the previous 30 days (e.g., for girls, “How often did you drink 4 or more standard glasses of alcohol on one occasion in the previous 30 days?”; for boys, the number of drinks was 5 or more). A figure showing different standard drinks was shown to make the concept more comprehensible [8,9,38].

For HID, we assess how many glasses of alcohol students drank each day during the last week. Based on this information, we estimated those girls who consumed 8 or more standard glasses of alcohol on at least one occasion, and for boys, the number of drinks is 10 or more [10].

For weekly consumption, we assess how many glasses of alcohol students drank each day during the last week. Based on this information, we calculate the total number of glasses consumed in the past week [8,9].

Any consumption is calculated using the question “On which days of the past week did you drink alcohol?” Possible answers were as follows: “Monday to Sunday”; “I haven’t drunk in the past week”; and “I have never drunk alcohol” [9].

#### 2.6.3. Risk Perception

We explore the perception of danger related to BD, as well as the seriousness of health problems related to BD (such as liver problems, alcoholism, or traffic accidents), and the probability of acquiring these problems. We use Likert scales with five answer options (from never to almost always).

#### 2.6.4. Attitude towards Binge Drinking

We explore four items measuring pros (e.g., “Drinking 4/5 or more glasses of alcohol helps me have fun with my friends”), and four items measuring cons (e.g., “I don’t like myself when I drink 4/5 or more glasses of alcohol”). We use Likert scales with five answer options (1 = absolutely disagree; 5 = absolutely agree).

#### 2.6.5. Social Influence. Model, Norms and Social Pressure

Social modeling is assessed by asking participants how often people in their environment (i.e., parents, siblings, (best) friend(s), girlfriend/boyfriend) drink alcohol and engage in BD (1 = never; 5 = always).

The social norm is measured for each person in their direct environment (i.e., parents, siblings, (best) friend(s), girlfriend/boyfriend) by one item “My (e.g., mother) thinks that …” 1 = “I am certainly not allowed to drink 4/5 glasses or more of alcohol” to 5 = “I am certainly allowed to drink 4/5 glasses or more of alcohol”.

Social pressure is assessed by “Have you ever felt pressure from (i.e., parents, siblings, (best) friend(s), girlfriend/boyfriend) to drink 4/5 or more glasses of alcohol?” We use a five-point scale (1 = never; 5 = always).

#### 2.6.6. Self-Efficacy

Self-efficacy is measured by ten items. Each item assesses whether participants feel able not to drink in a certain difficult situation (situations that would usually trigger BD, e.g., “How difficult or easy is it for you not to drink more than 3 (if you are female) or 4 (if you are male) glasses of alcohol if others around you drink 4/5 glasses or more of alcohol?”). We use a five-point scale (1 = very difficult; 5 = very easy).

#### 2.6.7. Intention

We use two questions about the intention of alcohol use and BD with a five-point scale (1 = absolutely will not; 5 = absolutely will), “Are you intending to generally reduce your drinking on one occasion (e.g., in a bar, at a party, etc.)” and “Are you intending to drink less than 4/5 glasses of alcohol in one occasion (e.g., in a bar, at a party, etc.)”.

#### 2.6.8. Process Evaluation

To assess the implementation compliance, the number of sessions performed by the participants are registered. After completing each session and in the post-test questionnaires, we ask respondents whether the intervention is useful, realistic, interesting, and personally relevant on a five-point Likert scale (e.g., 1 = Totally disagree; 5 = Totally agree).

In the final evaluation, we also assess the general satisfaction (e.g., 1 = Very unsatisfied; 5 = Very satisfied) and, by using a five-point Likert scale, the opportunity of learning, and the probability of using the counselling (e.g., 1 = Totally disagree; 5 = Totally agree) [56].

### 2.7. Procedures and Ethic Approval

The implementation of the intervention is carried out in a school context (a total of seven sessions during school hours) and it is the students who fill out the questionnaires. The first session in the different centers is guided through video conferencing programs (Google Meet, Zoom, or Skype) by a member of the project, who also facilitates website access and registration. Then, sessions 2, 3, 4, and 7 are conducted by the teacher at school, and sessions 5 and 8 are self-administered by the adolescents at home. Post-test evaluations or sessions 6 and 9 are again virtually guided by a member of the project.

In the ECs, the first session is carried out between October 2020 and February 2021 (there is a large timeline due to the COVID-19 pandemic). Then, the four following sessions are carried out in the subsequent period (every one or two weeks). The first follow-up questionnaire (post-test) is carried out from April to June 2021. Finally, the second challenge is carried out from September to December 2021, and the second follow-up questionnaire (post-test) is carried out between October 2021 and February 2022. Each session takes approximately an hour.

The CC only gets the three evaluation sessions: the baseline data, the first follow-up questionnaire (post-test) six months later, and the second follow-up questionnaire (post-test) twelve months later.

We endeavour to improve the implementation of the program through the following strategies [9]:Carrying out most of the sessions within the schools as part of the health promotion curriculum.Collaborating with teachers during the assessments through video conferencing programs (Google Meet, Zoom, or Skype) and phone calls, if necessary.Weekly contact with school counsellors to check the development of the program.Participation reminders are also sent via email where participants have not finished the intervention procedures, so they can complete them outside of school.User manuals for teachers and counselors to understand the website operation.

The interventions are carried out according to bioethical guidelines as formulated by the University of Seville: the students need to answer the questionnaires themselves and confidentiality is guaranteed, according to the General Data Protection Regulation (GDPR). Before initiating participation in the study, parental and individual informed consent must be obtained. The study had the approval of the Bioethical Committee of Andalusia, and the study was retrospectively registered at ClinicalTrial.Gov on 16 April 2021 (identifier NCT04853628).

### 2.8. Data Analyses

Differences between the conditions in the baseline sample are assessed using t-tests for continuous variables and chi-square tests for discrete variables. Moreover, when the dependent variable is not normally distributed, the Mann–Whitney U test is used.

Since pupils are nested within classes in the study, and classes are nested within schools, to examine predictors of dropout versus nondropout, a three-level mixed logistic regression analysis is conducted. To test the effectiveness of the program, we also perform three-level mixed logistic regression analyses for the outcomes of BD, consumption, and HID, and a three-level mixed linear regression for the outcome weekly consumption.

The independent variable is participating (EC-1, EC-2) versus not participating in the program (CC), and covariates are the outcome at baseline and sociodemographic variables. Moreover, the interaction effects between experimental conditions and all sociodemographic variables are entered as covariates into the analyses. To quantify the predictive power of the logistic regression models, Nagelkerke’s R2 is reported.

To study predictors of adherence, we also analyze the associations between potential participant characteristics (i.e., gender, age, academic course, religion, nationality, Apgar score, family affluence, and alcohol use at baseline) on the one hand and participation in the intervention (i.e., adherent or not) through the number of sessions attended by the participants at schools on the other.

Finally, for the process evaluation, a descriptive analysis is performed using chi-square tests to examine differences between males and females and between binge drinkers and non-binge drinkers. We use SPSS Statistics for Windows, version 26.0 (IBM Corp), for these analyses. The level of significance used for the main effects is alpha = 0.05, and alpha = 0.10 for the interaction effects. Effect sizes (odds ratios [ORs]) and 95% CIs are also calculated.

## 3. Discussion

Alcohol consumption during adolescence is a serious public health problem due to its numerous negative consequences, in addition to the psychosocial and community implications derived therefrom [2,4,57]. This paper describes the design, implementation, and evaluation of an animation- versus text-based CT game intervention aimed at reducing alcohol consumption, and specifically BD, in 14–19-year-old Spanish adolescents.

The development and implementation of (effective) programs for preventing alcohol drinking at schools represent an important issue as alcohol is the most consumed psychoactive substance among secondary education in Spain students between the ages of 14–18 [14]. Implementation often requires high levels of cooperation from educational centers which usually have temporary, structural, and organizational difficulties in providing the intervention, even more so in the context of the COVID-19 pandemic. Thus, one way to promote well-implemented health programs is to involve these educational centers and their stakeholders, especially teachers and counselors, and to receive effective schools-researchers cooperation due to its importance in the success and the acceptance of the programs [9,33,58]. In addition, the use of digital technology through delivering an online intervention, e.g., a web-based CT intervention, may improve the possibility of implementing these types of programs [29,31,59].

The main strength of this study is that it is based on the I-Change Model for predicting healthy behavior acquisition and is preceded by two previous interventions developed in the Netherlands [8,26,37] and Spain [9,34,38,39] with extensive research on the effectiveness and cost-effectiveness of patterns of alcohol consumption in adolescents. These previous studies revealed that a CT intervention could be presented in a very attractive, interactive way, like an animation-based game. Using animation and gamification elements to educate people about healthy behavior has been shown to increase motivation, cognizance, and understanding, and to change health behaviors [40,41,42,43,44,45,46,47]. Moreover, we carry out a long-term assessment to improve the effect on BD, and for the Spanish context there is no similar study targeting alcohol consumption using an animation-based CT game intervention in adolescents.

There may be limitations to the study. The biggest one could be attrition, which is a significant problem in CT interventions [8,9,26,34], either because of participants not using or not continuing to use the intervention (also referred to as low adherence to the intervention) or participants not completing follow-up measures (poor retention to follow-up) [60]. Jander et al. [8] obtained generally low adherence rates, being 21.39–67.31% at school and 0–1.66% at home, and a retention rate of 31.11% (*N* total = 824/2649). Lima et al. [9] attempted to improve on this, taking into the recommendations from this previous intervention developed in the Netherlands, and obtained adherence rates of 47.2–62.1% at school and 1.1–3.1% at home, and a retention rate of 50.92% (*N* total = 612/1247), which seems to show slightly better results.

To minimize attrition rates, and increase engagement as regards Alerta Alcohol 2.0, we follow different strategies. First, we design the intervention considering some of the recommendations of the original design after implementation [9]. Second, we introduce animated, video-delivered, tailored information which allows for more extensive information processing [42,43] and is related to reducing cognitive effort, improving understanding of the message, and thus increasing motivation for change [44,45,46,47]. Third, we incorporate new gamification elements to continue to encourage the user and take advantage of their maximum involvement [40,41]. Fourth, we carry out most of the interventions in the school environment [9]. Fifth, evaluation sessions (pre-test and post-test questionnaires) are performed in the virtual presence of a member of the research team [9]. Finally, reminders on participation are also sent via emails when participants have not finished the intervention procedures, so they can complete them out of school [61].

## 4. Conclusions

This study presents insights into the effectiveness of an intervention focused on reducing alcohol drinking and specifically BD in Spanish adolescents between 14 and 19 years of age. The findings will contribute to the development of future alcohol drinking pattern prevention interventions in adolescents. If the program proves to be effective, the ultimate goal would be regional and eventual national implementation.

## Figures and Tables

**Figure 1 ijerph-18-09978-f001:**
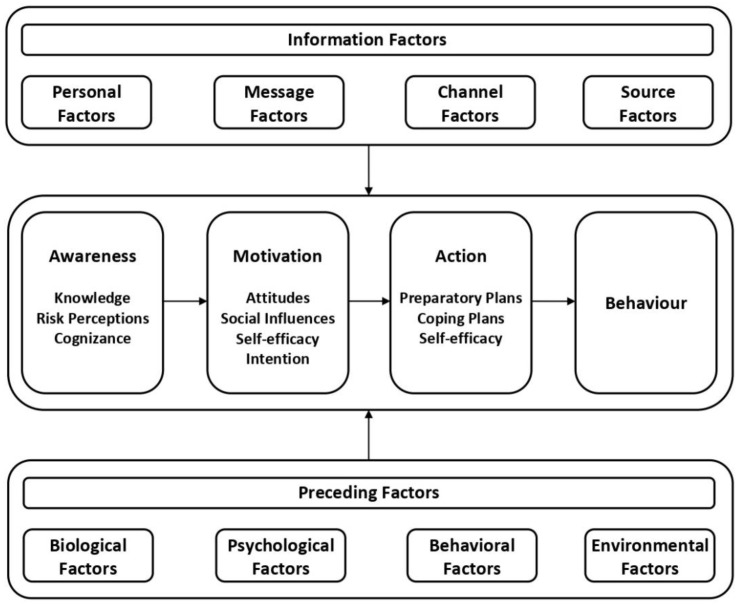
I-Change Model [18].

**Figure 2 ijerph-18-09978-f002:**
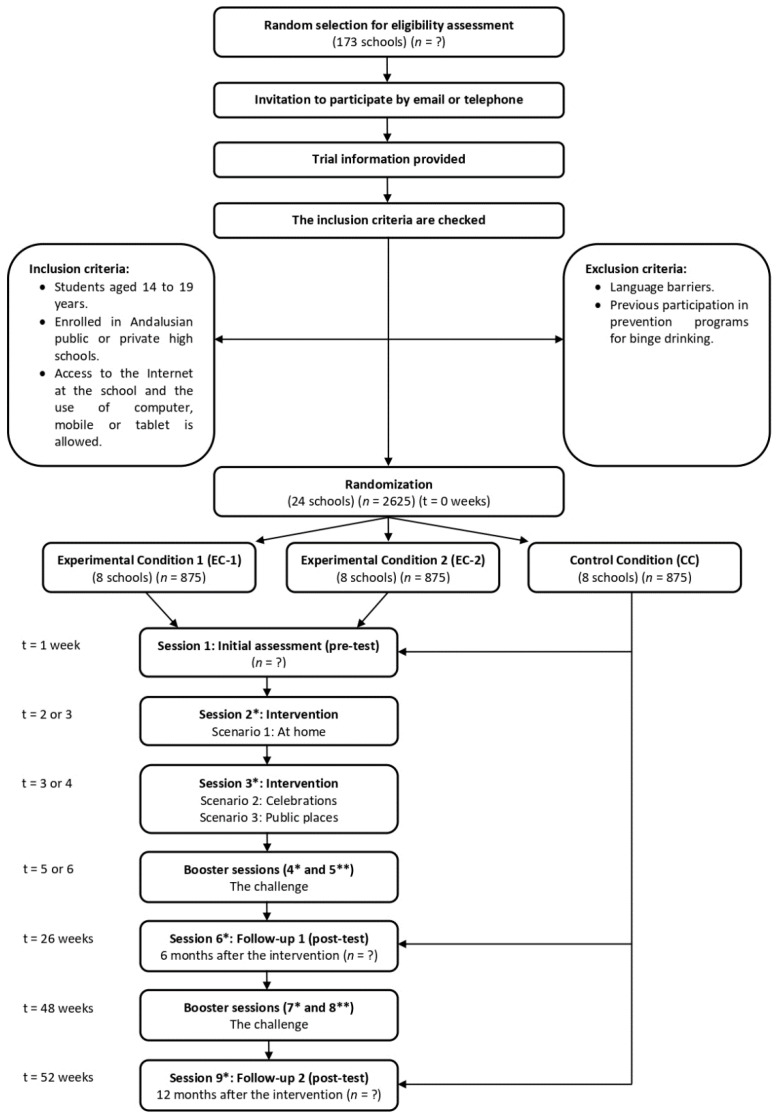
Flowchart of the intervention. * It takes place in the school environment. ** It takes place in the participant’s home.

**Figure 3 ijerph-18-09978-f003:**
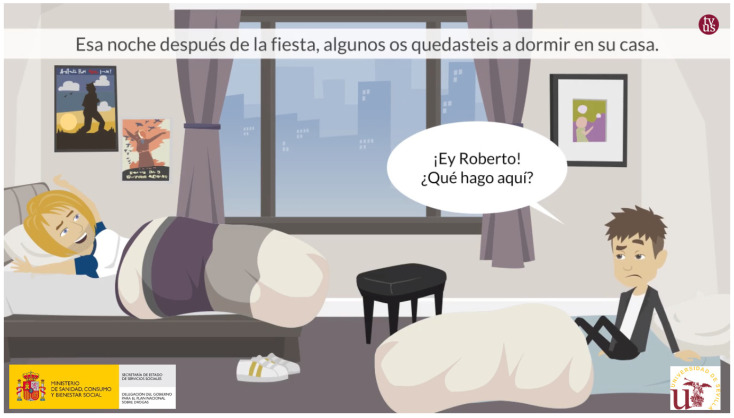
Screenshot of example story for a boy from the Alerta Alcohol 2.0 program.

**Figure 4 ijerph-18-09978-f004:**
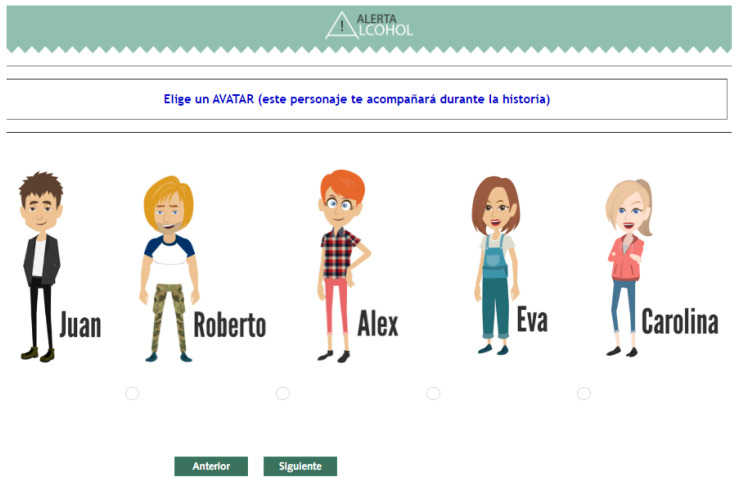
Screenshot of the page for choosing an avatar from the Alerta Alcohol 2.0 program.

**Figure 5 ijerph-18-09978-f005:**
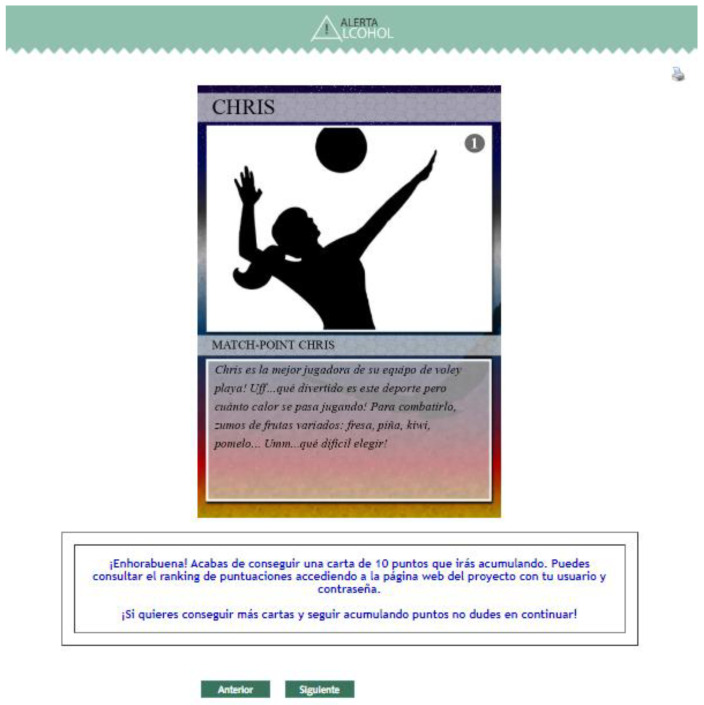
Screenshot of the page for receiving a card from the Alerta Alcohol 2.0 program.

**Figure 6 ijerph-18-09978-f006:**
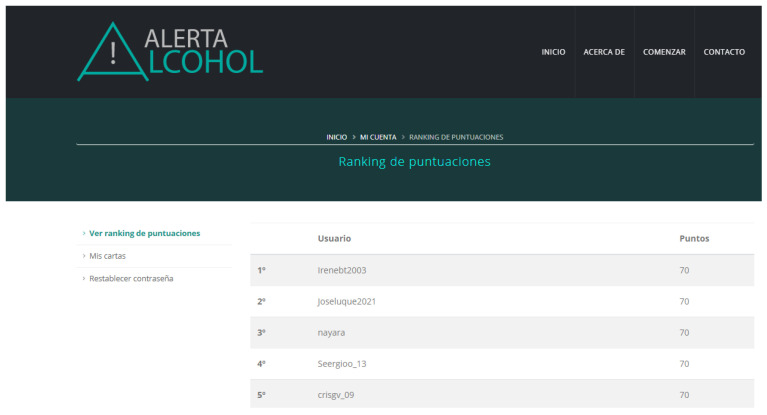
Screenshot of the page for ranking user position based on the accumulative score from the Alerta Alcohol 2.0 website.

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
