# Peer review of "An Animation- Versus Text-Based Computer-Tailored Game Intervention to Prevent Alcohol Consumption and Binge Drinking in Adolescents: Study Protocol"

_ijerph, 2021, doi:10.3390/ijerph18199978_

Round 1

Reviewer 1 Report

Study protocol of a Cluster-Randomized Controlled Trial to reduce binge drinking in adolescents through e-Health. A relevant issue is addressed with a well-founded intervention and using new technologies to adapt to the current context and the preferences of the target population. The trial aims to test the effectiveness of an improvement over a previous intervention. The main concern is related to the Methods section (missing information and instruments selected).

Some questions are suggested that could improve the manuscript (or the protocol if possible):

Abstract 

- Line 26: "primary outcome is BD within 30 days before evaluation". The wording is confusing. Consider this alternative wording: "primary outcome is BD within 30 days before post-test evaluations".

Keywords

- Consider substituting one of the keywords relating to alcohol use for Cluster-Randomized Controlled Trial

Materials and methods

- Since the protocol is presented and not the trial report, wouldn't it be more appropriate to use SPIRIT instead of CONSORT?

  • Flowchart: The n of those assessed for eligibility must be indicated. Consider using the Schedule Of Enrollment, Interventions, And Assessments of SPIRIT.

- Line 202. Not all possible centers being contacted? It seems that the centers are chosen in advance.

-Method used to generate the random allocation sequence must be indicated.

- Why aren't validated instruments used to measure consumption (AUDIT, TLFB, ...) and the other variables of interest? 

Minor considerations

  • Consider shortening some sentences that are too long (e.g. lines 35-42 or 466-469)
  • Line 165. A parenthesis is missing

Reviewer 2 Report

Lima-Serrano and colleagues provide a protocol for a cluster-randomised controlled trial evaluating the effectiveness of two versions of a computer-tailed technology based intervention to decrease binge and high-intensity drinking in Spanish youth. The two interventions, a text-based and an animation-based intervention, will be contrasted against a no-intervention control.

I found the design of the intervention and the planned analyses to be, on the whole, sound. I highlight a few questions below which remain after reviewing the protocol.

Main concerns

Would the authors be able to place their drinking behaviour measures in the context of the wider alcohol consumption/drinking behaviour screening and detection literature? These appear to be novel measures. Is there a reason that existing instruments such as the AUDIT, AUDIT-C or ASSIST are not adapted or used? If the authors are using questions that have been shown to be sensitive and specific to detecting BD and HID in adolescents then please state so and provide these values. If not, is there a reason not to use existing instrument that have been used in adolescent populations?

Similarly, the assessment of the amount of alcohol consumed in the past week (lines 332-334) appears to be akin to the Timeline Follow Back procedure that is commonly used in alcohol consumption assessment. Is this the case, or is this procedure different?

The measures described in 2.6.4 through 2.6.8 all appear to use a five-point Likert scale except for the social modelling item, which uses a four-point scale. Are these existing measures that have been validated or have they been constructed specifically for this trial? If the former, please provide citations and relevant statistics demonstrating reliability and validity. If the latter, is there a reason that only one scale uses a four-point Likert? There is nothing inherently wrong with this, it merely piqued my curiosity.

How will adherence or not (line 432) be determined?

The authors propose a .10 alpha for their interaction terms. What is the justification for this?

The main effects alpha term is missing (line 436).

The authors do not provide plan for how to deal with differences in the baseline, if any emerge. Also, what is the plan for uneven attrition between conditions, should this arise?

Do the authors propose to formally evaluate the I-Change Model? It is not clear from their analysis plan if this is an aim of the work.

Minor comments

The sentence beginning line 39 is confusingly worded.

The authors do not define what is a standard drink unit in Spain. This can differ based on region. In the USA it is 14 grams, while in the UK it is 8 grams of pure ethanol.

The sentence beginning line 57 requires citations.

The prevalence data provided on line 64, why do the authors provide 12 month and 30 day prevalences, making it difficult to extrapolation the two.

Line 66 is the first time the authors mention Andalusia without specifying that it is a southern region of Spain. This is mentioned much later in the manuscript.

Line 70, the comma after 'however' should be a period.

Line 80, the word 'model' is omitted after I-Change.

Lines 93-95, how effective is the model in predicting behaviour and how useful is it in preventing alcohol consumption? Examples would be helpful here to motivate the use of the model.

Similarly, line 117, how effective was the intervention in reducing BD? What was the drop and how long was it followed-up? Same question for lines 126-128.

Lines 142 and 145, user' should be users'.

Figure 2, student's should be students'.

Reviewer 3 Report

It is an honor to review this exciting topic of research on the prevention of adolescent drinking. The study has an interesting topic and is overall well presented in this paper. The opinions that I would like to add or modify in this research protocol are as follows.

1. Please describe whether the research team has sufficient and appropriate expertise and experience in developing and operating the program.

2. The study endpoint(s), contents of program, and study variables clearly stated and are acceptable. 
For sessions 2-5, at what interval and for how long does each session last?

3. Describe in more detail what happens in the booster session 7 and 8.

4. Describe in detail how consent will be obtained from students and parents.

5. As a matter to minimize bias, the randomization method used for assignment of participants to treatment groups, allocation to treatment groups concealed, delivering treatment blind to treatment assignment, outcomes assessors blind to treatment assignment should be clearly described.

Round 2

Reviewer 2 Report

I thank the authors for their thorough consideration of my review and am satisfied with their responses.

I look forward to reading the outcome of their intervention.